

# A comprehensive secure system enabling healthcare 5.0 using federated learning, intrusion detection and blockchain

Jameel Almalki[1], Saeed M. Alshahrani[2] and Nayyar Ahmed Khan[2]

[1] Department of Computer Science, College of Computer in Al-Lith, Umm Al-Qura University, Makkah, Makkah, Saudi Arabia
[2] Department of Computer Science, College of Computing and Information Technology, Shaqra University, Shaqra, Riyadh, Saudi Arabia

## ABSTRACT

Recently, the use of the Internet of Medical Things (IoMT) has gained popularity across various sections of the health sector. The historical security risks of IoMT devices themselves and the data flowing from them are major concerns. Deploying many devices, sensors, services, and networks that connect the IoMT systems is gaining popularity. This study focuses on identifying the use of blockchain in innovative healthcare units empowered by federated learning. A collective use of blockchain with intrusion detection management (IDM) is beneficial to detect and prevent malicious activity across the storage nodes. Data accumulated at a centralized storage node is analyzed with the help of machine learning algorithms to diagnose disease and allow appropriate medication to be prescribed by a medical healthcare professional. The model proposed in this study focuses on the effective use of such models for healthcare monitoring. The amalgamation of federated learning and the proposed model makes it possible to reach 93.89 percent accuracy for disease analysis and addiction. Further, intrusion detection ensures a success rate of 97.13 percent in this study.

## INTRODUCTION

Artificial Intelligence (AI) is gaining popularity across all the sectors. Healthcare units are finding their ways in the use of machine learning (ML) and associated algorithms. An emerging technology in the healthcare unit is the Internet of Medical Things (IoMT) (*Almalki et al., 2022*). Blockchain and machine learning have a significant role and made a significant contribution to various systems over the past decade. These capabilities are enhanced with the help of AI in healthcare and precisely, it is termed competent healthcare. AI and ML-enabled devices and technologies not only empower the process of prediction and management of patient data, but also reduce the healthcare cost and stress of the patient. The more significant challenge is to identify all the data flowing from these devices and collect it at a single place. While the data is collected and collaborated at a single storage location with the help of machine learning algorithms, but the primary issue that arises is privacy and security. The compliance of the data and ownership concentration is

Corresponding author
Nayyar Ahmed Khan,
nayyar@su.edu.sa

indeed one of the most important factors under consideration during modern research. Federated learning (FL) is very helpful in providing a centralized model for solving the issues arising from ownership and security. Several objects, such as sensors or smart devices, are connected in the healthcare setting to monitor observations. The data collected from these objects related to the human body and its apparent mechanisms is restored for further analysis and prediction of diseases. Smart hospitals are now replacing the traditional medication policies for patient observation. The improvement in sustainable smart cities is affecting the day-to-day life of humans. One of the most challenging and developing new technology perspectives for smart hospitals is using IoT-oriented structures for monitoring patient health (*Karatas et al., 2022*) called the Internet of Medical Things (IoMT). Public safety, as well as improvement in healthcare, is one of the important factors that relates to the use of technology with the help of the Internet of Things (IoT) and IoMT. With the help of such techniques, any individual can identify and measure information related to their persona during a healthcare crisis. Just by connecting to the Internet and observing the information flowing from the IoMT devices, healthcare professionals can now monitor the health of any patient from remote locations (*Sujith et al., 2022*). The real-time identification of data and proper diagnosis is one of the most critical factors achieved through these technologies. The information collected by the devices in IoMT is used for the early identification and accurate diagnosis (*Ogundokun et al., 2023*), especially for those patients who face difficulties in visiting the hospital. Further there are various challenges faced by patients such as old health, time of travel, contamination of samples, large queues at the hospital sites, availability of resources in the hospital, appointment delays, polluted environment, and availability of specialist healthcare professionals (*Heydari et al., 2022*). Using on-premises healthcare facilities is gaining popularity for the reasons described above. The medical examinations are also done at the patient's convenience at their houses by various pathological labs and hospitals. The setup of an intelligent monitoring system for the patient's health can be designed to identify the observations and collect the data for diagnosis by the doctors. This technology connects and collaborates with the patients and the physicians to monitor the health indices and predict the patient's health issues. The on-premises healthcare facility comprises devices that can monitor the oxygen level, BP rate, body temperature, heartbeat rate, ECG values, and many more observation sets (*Makina & Ben Letaifa, 2023*). The data collected from the system is sent to the server with the help of several IoMT devices across wireless networks. These information are of immense importance for the medical professional for the purpose of diagnosis.

*Razdan & Sharma (2022)* reported many use cases where IoMT is integrated with modern healthcare to provide better quality prediction and diagnosis of diseases. The prediction and diagnosis of disease with the data collected from these devices carried out using sophisticated machine learning algorithms (*Baduge et al., 2022*). *Ahmed et al. (2020)* acknowledged the use of artificial intelligence related algorithms in several medical healthcare units. These algorithms can identify and discover the disease that a patient might be suffering from. The medical treatment and the support needed for the patient are also protected along with the disease prediction in this case. The use of IoT for bio-research and bio-manufacturing was explained by *Shamhan et al. (2023)*. ML applies to various sectors

across human life, which include image processing, computer vision, augmented reality, natural language processing, and healthcare. However, machine learning algorithms can predict accurate results with the help of an extensive training sets. Predicting precision is one of the most critical factors that must be reached when a machine learning algorithm is applied to a training data set. Critical observations from various IoMT devices are collected on centralized storage where the machine learning algorithm tries to predict a patient's health status (*Abbas et al., 2023*). To overcome the precision problem, a bulk volume of data is analyzed and stored on a centralized cloud services. These data are analyzed, and the health condition is predicted depending on the various characterization and parameters under consideration. However, the privacy problem for patients data remains essential in the medical field. Several models are presented by various authors to support the problems associated with the IoMT and healthcare. The core idea is to use technology so that disease diagnosis and predictions becomes simple. This marked the beginning of a new era of computational healthcare in which IoMT plays and important role.

## Blockchain

An exciting technology that can help to deliver trust in patient privacy is blockchain technology (*Ashfaq et al., 2022*). The critical issue of security and privacy, which relates to data protection in the healthcare units, is resolved with the help of integrating blockchain into innovative healthcare units. Several cyber security architectures are now using blockchain technology to ensure that data stored in a cloud service is not compromised. It is beneficial for patient records to be available electronically to allow the distribution of information to healthcare professionals and the development of training sets for use in machine learning models. This idea has led to a drastic increase in the use of intelligent healthcare mechanisms (*Singh et al., 2022*).

Blockchain is gaining importance due to the decentralized data storage and privacy protection it affords. The use of blockchain has been encouraged by various researchers in their models to provide privacy and transparency among transactions that take place across the network. Peer-to-peer (P2P) networks collaborating in the blockchain contain sensitive information related to connected IoMT devices and sensors. This information can be supported and preserved with digital encryption to ensure privacy. In various proposed models, the concern of privacy is a major issue where the customer or user finds it difficult to trust the network. The information travelling across the blockchain network is safe and secured due to information exchange based on secure digital key exchanges. A hybrid blockchain has been suggested as a mechanism to be used in a model (*Alangari et al., 2022*). The secure key exchange takes place and, finally, the system transfers the data across the P2P network.

Various problems are associated with the security and privacy of the data in the IoMT (*Taherdoost, 2023*). Combined networks incentivize hackers to find information in the connected devices. The malicious purposes of hackers might be to find valuable data across these networks and their peer nodes. The capturing of medical records and information of individual patients from healthcare devices and servers is a significant threat (*Hireche, Mansouri & Pathan, 2022*). All the devices connected to the network are

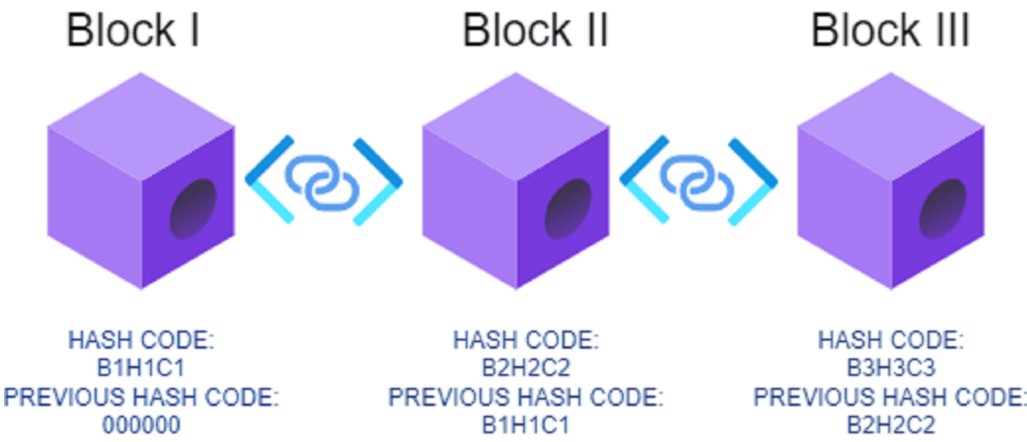

**Figure 1** Demonstrative blockchain architecture for security and privacy.

integral to data collection, which can be used for any malicious reason. It is also possible that phishing for and spamming of information can be done with the help of simple hacks by unauthorized access to these networks (*Roobini et al., 2022*). The devices that are a part of intelligent healthcare units are connected directly to wireless networks without any security key. The information that flows from these devices can be easily captured and might lead to problems associated with networks such as distributed denial of service (DDoS). It is also feasible that the data can be tempered and modified without the notice of the units taking care of the information. False reports can be generated with the help of incorrect information modified by hackers. The previous issue arose because of the centralized information storage in the IoMT realm. Record forgery, along with the manipulations of the information, altering the device configurations, and attacks on the devices as gateways to the server, can take place quickly.

## Security and privacy

The stated problems and issues can be resolved with the help of blockchain-based techniques on the computer networks under consideration (*Pathmudi et al., 2023*). Several independent networks in blockchain are the carrier for sending information from one place to another (*Rahman et al., 2023*). As per Fig. 1, the architecture for using blockchain can be understood as using cryptography techniques to generate security keys for data transmission. The sequential blocks are cryptographically secured with the help of public and private keys. The information is centralized in its union, and it is transferred from one place to another with the help of a hash code. The hash value is constant and cannot be altered. It also provides the decentralization of the data (*Ogundokun et al., 2022*). Along with this, blockchain is very helpful to provide transparency in front of the users trying to access the data from the blockchain. Despite being transparent and resolving privacy issues, blockchain implementations in the present state still have drawbacks that need to be answered (*Díaz & García, 2023*).

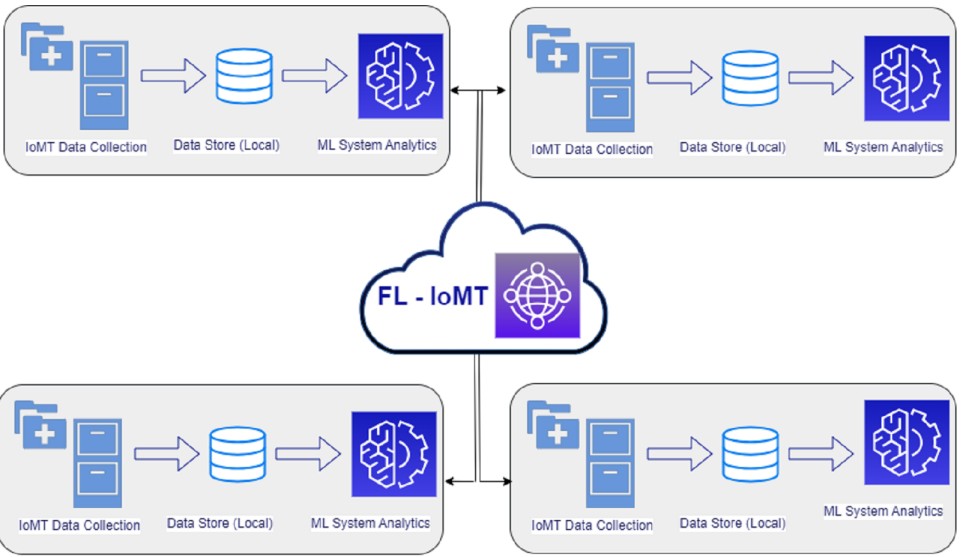

**Figure 2** Federated learning for ensuring data security and confidentiality in a smart healthcare unit.

## Federated learning

To overcome problems associated with blockchain and to provide a more sophisticated mechanism for identifying diseases in the proposed study, federated learning (FL) can be effective (*Ogundokun et al., 2022*). Several devices collaborate their machine learning principles without sharing the data in FL. In general, FL is used in places where only the derived rules are shared without sharing the information. The data collected from various healthcare unit are supplied to machine learning algorithms, and the results are extracted. Several healthcare units share their learning with a pool of information. Irrespective of the data analyzed, only the results obtained are supplied to the FL zone (*Li, Luo & Cao, 2022*).

Figure 2 represents an FL-based smart unit for healthcare. In this diagram, individual healthcare units prepare their local model to train the data collected at particular locations. The centralized data for the local healthcare unit is loaded into the regional training model. The results gathered from the local training model are uploaded to the centralized global model. The data is aggregated in the worldwide model from various individual units. Finally, the global model is applied to predict disease from the local training uploaded by different healthcare units. It is worth mentioning that no actual data is transmitted in this model. FL is a famous technique to ensure the safety of information (*Saraswat et al., 2022*). Instead of accumulating the data from various sources and then applying the machine learning model, the training is done for a standard global model on an individual basis where the data division occurs at the relevant organizations. The training data across the units is collected and supplied from the sources to deploy a global model in which the actual datasets are not exchanged, thereby maintaining the privacy and security of individual data.

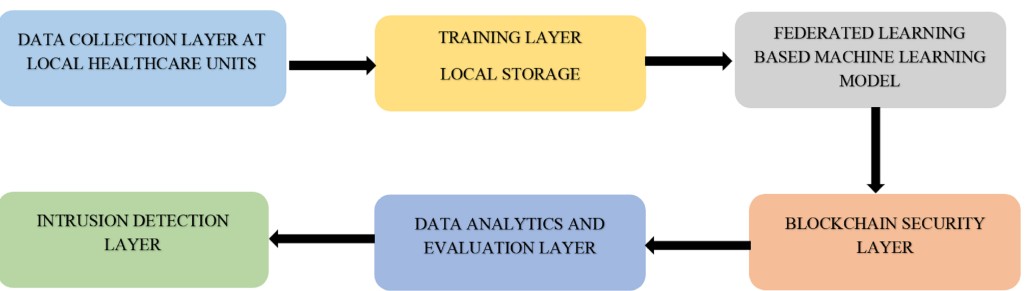

**Figure 3** **General workflow of the proposed model in this study.**

## Healthcare 5.0—fifth generation

A recent paradigm in healthcare is Healthcare 5.0, which can be deployed to provide various healthcare services (*Taloba et al., 2023*). These services are provided by utilizing new information technologies, such as the (IoT), (AI), big data analytics, blockchain, and cloud computing. The network architecture uses the latest communication techniques responsible for linking various healthcare devices and services. The data is created with the help of sensor-based devices and analyzed with the help of AI techniques, stored in centralized cloud storage services, and used for the prediction and analysis of patient information. Several problems were associated with Healthcare 4.0 related to the transfer of information (*Kumar et al., 2022*). The use of automation, along with sophisticated machine learning algorithms in the newer version, provides emerging technologies. The monitoring of the patients, along with the remote diagnosis, is made possible with the help of this technology in a more straightforward and effective manner. Various AI applications find their way across this technique for predicting diseases based on image processing (*Zhang et al., 2022*).

With the advancement and improvement in the techniques used in association with AI, the complete realm of Healthcare 5.0 can be termed a smart healthcare system. Various solutions are proposed by different researchers and studies that make use of such techniques to empower competent healthcare and motivate the use of intelligent diagnosis systems empowered by AI and machine learning, as shown in Fig. 3. The IoT devices enables the collection of data in a very sophisticated and more straightforward way (*Gosselin et al., 2022*). However, there are certain loopholes in the privacy and security of the data. To overcome these issues, use of FL and blockchain is indeed very helpful (*Rajasekaran et al., 2022*).

Several problems are concerned with such type of architecture. Confidential clinical data can be compromised while transferring it to the blockchain (*Borkovcová, Černá & Sokolová, 2022*). This is indeed a critical issue related to security and privacy. Incorrect readings and appropriate data monitoring may also be noticed due to the negligence of human interventions or inaccurate data recording. The medical devices recording the observations may also malfunction, resulting in data not being recorded accurately. If the information recorded is inaccurate, then the information processing to FL might be

compromised and take the wrong shape. Several researchers have proposed a variety of models to solve the problems. Figure 3 below represents a schema for this study.

Several contributions are made in this article, some of them are as follows:

1. The model created in this study comprises FL and blockchain infrastructure to provide a collaborative model that securely records information and manages training for the healthcare framework.
2. A higher level of security is assured with the help of blockchain so that the processed information is not compromised at any stage. We also present FL integration to ensure that no data is lost or compromised during the training process.
3. The integration of smaller units and the resultant training model is beneficial to generate the final macro-model for medical organizations.
4. The integration of FL enhances the training process of data available in the local model to a very high level. The computational procedure for preparing the local data will be quick, providing practical information and training rules.
5. The detection of intrusion at the transfer level in our study comprises the identification of any attack from hackers or unauthorized access patterns.

Thereby, it will increase the security features of Healthcare 5.0 to a considerable extent. The system's reliability with the integration of FL and the healthcare units provides proper safety and secrecy of the information. The data is safely processed and securely distributed over the network, saving sensitive information for every patient. With the help of minimal resources, all the information is secured effectively without failure. The main ideology behind this research is to find and develop a model based on the Actual Time—Deep Learning Model (AT-DLM) for identifying diseases and detecting intrusion at any level with the highest accuracy.

The rest of the article is organized as follows 'Literature Review' reviews relevant research to the developed model. 'Methodology' provides the methodology that is adapted during the development of the model in this study. 'Simulation Results' identifies the simulation and the practical demonstration for calculating the findings. 'Conclusion' finally discusses the results and the conclusion for the proposed model.

## LITERATURE REVIEW

Several studies have identified the use of blockchain in various processes. One critical processes is the healthcare unit where blockchain technology has found an excellent level of use. The integration of information from multiple sensors and other devices in healthcare, along with data distribution across various networks, is explained progressively by *Jin et al. (2023)*. The analysis of the applications recommended in multiple P2P networks for information sharing are explained in depth. The deployment, as well as the ability for various blockchain-based units including intelligent hospitals, smart cities, analytics of big data from multiple departments, integration of AI and payment gateway services are some of the well-known use cases for the integration of blockchain technology. However, a study (*Jin et al., 2023*) concludes that the impact of blockchain technology is still set to expand across the planning of the various security features for smart cities and smart homes.

Devices connected to the Internet in the form of sensors or IoT-based units lack security and safety (*Hannan et al., 2022*). Safety and security are critical in many sectors including the automobile industry where the use of blockchain technology is expanding (*Du et al., 2021*). *Ihnaini et al. (2021)* examined the factors that affect the adoption of blockchain technology in healthcare units including all the issues that are associated with intelligent healthcare. The solution was proposed with the help of a blockchain and the development of an application-helpful framework for innovative healthcare units. Yet another exciting model for predicting diabetes diseases was proposed by *Khan (2021)*.

*El Kafhali & Salah (2019)* suggested an analytical model based on a network of queues with the ability to estimate the minimum required number of computing resources to meet a service level agreement. *Dhasarathan et al. (2023)* proposed a model that was capable of handling a vast amount of IoMT data by adopting blockchain for IoMT to preserve sensitive clinical information. *El Kafhali, Salah & Alla (2018)* suggested a good mechanism for how to reduce the cost of computing resources while guaranteeing health request performance constraints, particularly response time when accessing medical data stored in a fog-cloud environment. *Ai et al. (2021)* reported state-of-the-art methods related to intelligent healthcare and rehabilitation, including developments, methodologies, and applications to biological sensors, IoT infrastructure, soft robotics, bio-mechanical modeling, novel algorithms, big data analysis, and devices and tools for health monitoring. *Yang et al. (2020)* developed a smart and interconnected system for cardiac health management in the context of a large-scale IoT network. *Selvakanmani & Sumathi (2021)* reported a robust mechanism using fog computing in association with cloud computing to enable a secure healthcare model.

The theories and models developed by various researchers highlights the need for a more robust and secure model to enable authentic healthcare. The method used for prediction comprises a machine learning algorithm and the fusion of deep learning. Improvement in the efficiency of the system, which is used for the prediction of diabetes and recommending all the possible solutions at the time of threatening conditions for life, was done with the proposed model.

In the present era, information can flow from various nodes across the network irrespective of specialized dedicated channels for transmission. Medical information flows from multiple domains and devices with the help of the service provider channel. IoT devices are indeed one of the most important sources of this information. Sophisticated sensors, wearable devices, image processing-based machines, intelligent monitoring units, *etc.*, are used to identify various body ailments of a patient. A massive amount of information is collected with the help of these techniques, which is managed and organized in a pool of data storage at a centralized server-based location. The data collected by these devices and stored in the centralized unit is used for identifying and predicting diseases (*Khan et al., 2021*). For the old, aged people, a monitoring framework was suggested by *Khan (2022)* in which the actual problems and needs were identified with the help of monitoring techniques. Machine learning was introduced to help overcome the issues associated with the old, aged people in this case.

*Alsulami et al. (2021)* demonstrated the use of federated learning in bio-medicines. Several solutions were supported and suggested for exclusive problems and challenges related to privacy concerns. Federated learning has delivered the proper safety needed for the information, which was immensely important for medical healthcare units. Going on further, *Xu et al. (2021)* revealed the use of machine learning in bio-informatics, in which the information was processed and managed using visualization techniques. The review that was presented also comprises information about various countries that are taking care of healthcare research and identifying the significant difficulties associated with objectives for machine learning in the healthcare industry.

The identification and prediction of breast cancer strategy with the help of data fusion in a secure deep learning environment was proposed by *Li et al. (2021)*. The model's accuracy was further enhanced by integrating a proper decision-based system with the help of deep learning algorithms. The data fusion approach was well-defined and enhanced for improved healthcare monitoring by *Mihaylov, Nisheva & Vassilev (2019)*. The entire control of the system was managed with the help of sensors. The use of AI for the diagnosis as well as protection of the treatment is also becoming popular recently. *Qi et al. (2020)* suggested a mechanism for the forecasting hospitalization of a patient into critical conditions with the help of AI. This study provided a classification model using significant input parameters for possible savings from medical expenses in healthcare. The compliance of several types of drugs in the procedure adopted towards medical diagnosis for heart problems was suggested by *Lorenzoni et al. (2019)* with the help of support vector machines (SVM).

The intensity of the medical ailment during the infection caused by COVID-19 was also addressed with the help of an AI scheme by *Wang et al. (2020)*. The research also focused primarily on a prediction of medication using historical information about the patient. The identification of the infection by coronavirus was also done with the help of a convolution neural network (CNN) by *Tariq et al. (2021)*. The model that was proposed by *Sedik et al. (2020)* makes use of a federated learning environment technique to identify and process the data collected over edge devices from various nodes by creating their confidentiality factors. The security of the data collected from multiple nodes of a similar healthcare facility was handled in the proposed architecture. Yet another prediction model was proposed by *Chang, Fang & Sun (2021)* for identifying various diseases related to cardiovascular issues with the help of federated learning. This model comprises the integration of Federated learning with support vector machines (SVMs) and solves all the problems that were noticed during the integration.

## METHODOLOGY

The model proposed in this study is represented in Fig. 4 with details and explanation follows.

### Blockchain component

In 2008, Satoshi Nakamoto developed the first-ever decentralized blockchain that could transfer data with secrecy and security (*Lorenzoni et al., 2019*). The entire block shared as a data block comprises an initial value of the blockchain block ID and the previous block's

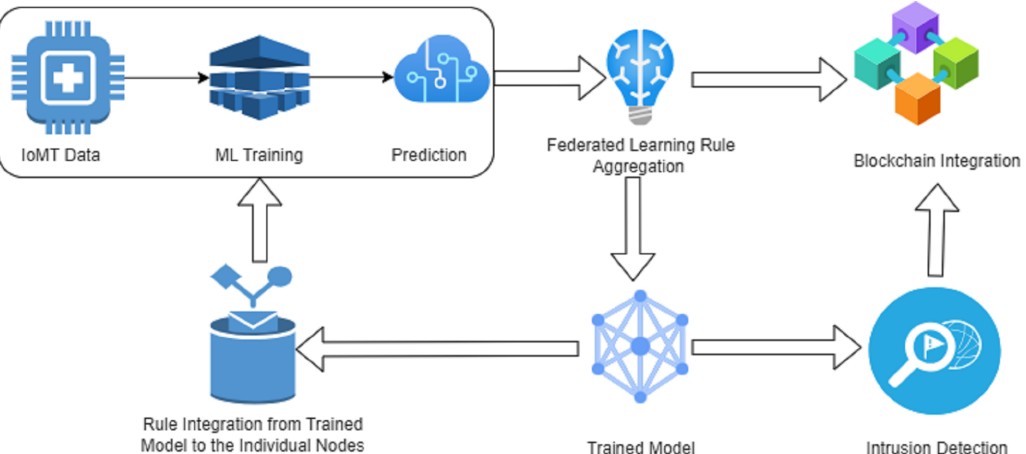

**Figure 4** **Proposed framework for IoMT enabled fifth-generation healthcare.**

hash code. Other information, such as the transaction and nonce at various timestamps related to the trades, were also transferred along with the existing block. Initially, anyone accessing the block tries to identify the hash value for that block. The information becomes visible whenever the hash function delivers the exact value of the hash code. Initially, the existing block must be searched, and after the search is over, a new block is identified. For an eligible block value with proper hash code, the contents are displayed, which remain unaltered and are in read-only format. This is the core of blockchain technology in which the information, once committed to a block, along with a proper hash code value, is integrated and remains unaltered. In the case of the intelligent healthcare units that are an integral part of this study, where the information in the block communicates with the centralized storage of data units. The data collected from several nodes in different networks and IoT devices is restored as a blockchain framework. All the transactions inside these blocks are stored for further transfer. The hash value for every partnership is unique and can only be determined once and when the proper decoding mechanism is applied. Even after the instrument is applied and the contents are retrieved, discontent remains unaltered due to its read-only nature. Merkle tree is used to identify the contents for revision and analysis in various transactions in the blockchain network. The versatile nature of the blockchain makes it compatible with the fifth generation of healthcare systems that comprises the IoMT computational paradigm.

## Actual Time—Deep Learning Model (AT-DLM) component

The data collected from various IoT devices and sensor-based monitoring services is centralized for further analysis and prediction. The Actual Time data contains the information that flows from the sensor-based devices and monitoring services to diagnose health ailments and predict disease. The consumption of patient energy, the requirement for various inventory services, and the management of multiple operations for transportation are several applications that are also considered in this framework (*Nakamoto, 2008*). The AT-DLM methodology is conducive to eradicating mistakes if errors arise in the

observation and monitoring. The approach to predicting and diagnosing the disease is one of the crucial factors for this implementation. The main target for the entire study is based on the evolution of a plane model, which is responsible for forecasting diseases and ailments.

## Federated learning component

Federated learning (FL) is a very new computational paradigm in which the safety and security of sensitive data, which is critical for any process, is ensured. The data fragmentation takes place at the local terminals, and finally, it is analyzed to generate the rules for the main module. In place of a combination of the data from various sources to generate a considerable dataset, this technique analyses the data at their respective organizational units. It makes use of the analyzed data to train the global module. In a way, it uses the divide and conquer policy to analyze the data from various smaller units locally. The training rules are forwarded to the centralized global team for further analysis of the large data blocks. It ensures the confidentiality of the critical data without transferring a single bit to a centralized server location or a cloud service platform.

Various organizations try to implement FL in which a centralized server holds all the rules after the training of individual units is completed. As a result, instead of the volume of this data at the centralized server location, only the train of the data model resides. The actual deep learning model, derived from various training data set rules, is integrated at the server level. This model handles the data that flows across the complete server unit. The server becomes a module trained by receiving the training rules from various remote locations at different data centers that might include a variety of medical healthcare units or hospitals. The training process at the individual units uses the data that IoT-based devices or sensor units contribute to the local repository or server. It ensures that the data does not flow across the network, which might hinder or compromise critical information. The training gradient of all the modules are integrated and applied in the final deployment of the server model. The entire set of rules are modified at different levels after taking feedback from various participants. The excellence of the data module implemented at the server level depends upon the close analysis of the training set rules derived from multiple local nodes. The feedback from various places is considered uniquely since various regulations are altered. The development of the model takes place in numerous iterations and teh prototype modules. These iterations help to generate a profound global model after performing the FL cycle innumerable times. Federated learning thus plays a very vital role in the proposed study to ensure that data is committed from different locations in a proper manner.

## Data amalgamation

In this study, data is collected with the help of various devices connected to different healthcare units. These devices are sensor-based or monitoring units integrated with the patient body or surroundings. The extraction of the data takes place from these devices, and the data flows across a secured framework in the local network. Once the data is collected at the local node, it is stored in a storage repository. The security of all the sensor-based

units is one of the significant factors considered in this study. The data collected from these units is challenging to handle across a substantial region. To manage the efficacy of the proposed model, the amalgamation of the data from various sensor units is done in such a way that provides proper results. The IoT devices usually comprise sensor-based units to evaluate the fifth generation of competent healthcare. The data which is collected by these devices needs to be processed. For example, the blood pressure measured with the help of an IoT device is combined with the oxygen level in a patient's body. The observation should go in synchronization with the sensor that measures the cardiac rate of a human being. The main application layer in this proposed system includes many sensor-based devices, wearable units, and monitoring systems. The collection of data is an automated process and thus the data collection from several IoMT devices is expected to occur in a seamless way from the local networks. However the expectation from the networks is to provide a malware free environment for data collection.

## Intrusion detection component

The detection of unauthorized access in the fifth generation of the healthcare system is one of the most essential *Yu et al. (2015)*. The data collected for final analysis at the local nodes in the healthcare units makes use of the deep learning model. Any intrusion trend observed in the data flow framework for the proposed system is monitored. For the blockchain-based application, it becomes practically simpler to handle the intrusion traits (*Alshahrani et al., 2022*). To develop an algorithm for the management of such applications (*Tavallaee et al., 2009*) suggested the intrusion detection mechanism. The security of the centralized server-based units can be managed without hesitation with the proposed framework's help. Deep learning can be employed in an intelligent way for a more effective intrusion management and detection system, as in Fig. 3. The central idea proposed in this study is to provide a blockchain-driven framework capable of disease prediction and empowered by FL for the fifth generation of healthcare systems. There are several components that work for the integration for intrusion detection in the model proposed. Data amalgamation from various sensors travels across the local network and gets collected in the data analysis module. However, the use of intrusion detection techniques ensures that the components are from authentic means and there is no threat to the privacy of the individual.

## Dataset component

For the prediction of the diseases and the proper diagnostic of the model proposed in this study, the Parkinson's dataset is used to detect intrusion, as shown in Table 1. The publicly available dataset NSL-KDD is considered for testing this model (*Lee et al., 2020*). Approximately 41 associated rules per record are generated in this dataset, with a proper declaration and definition of the entire information. The main objective for using this dataset under the study is to identify the difference between a healthy person and an individual suffering from Parkinson's disease. The two determining flag values for being healthy and non-healthy are taken care of with the help of the "status" attribute".

All the features in the Table 1. represent various values responsible for identifying the prediction and diagnostic properties of Parkinson's disease. Based on the observations

**Table 1   List of features contained in dataset of Parkinson's disease.**

| Value | Feature | Value | Feature | Value | Feature |
|---|---|---|---|---|---|
| 1 | Fo (Hz) | 9 | Shimmer | 17 | RPDE |
| 2 | Fhi (Hz) | 10 | Shimmer (db) | 18 | DFA |
| 3 | Flo (Hz) | 11 | APQ3 | 19 | Spread1 |
| 4 | Jitter% | 12 | APQ5 | 20 | Spread2 |
| 5 | JitterAbs | 13 | APO | 21 | D2 |
| 6 | RAP | 14 | DDA | 22 | PPE |
| 7 | PPO | 15 | NHR | 23 | Class |
| 8 | DDP | 16 | HNR | – | – |

taken from this dataset, the model proposed in this study's architecture is implemented to detect the disease.

## Working model component

Federated learning is integrated into this study, using the AT-DLM to provide value-added results. The intelligent responses collected after the application of federated learning are transferred to the blockchain to increase the security level. To improve the understanding of the algorithm used in this case for predicting and diagnosing the disease, further information can be added as given by *Ihnaini et al. (2021)*. The integration of the blockchain for the data to be transferred from one node to another in the network is recognized with the help of the platform suggested by *Arpaci et al. (2021)*. We have closely examined the deployment of the DLM technique used in this study. The use of sensors, monitoring units, and movable devices, along with the IoMT systems, are increasing daily to offer more intelligent fifth-generation healthcare. Innovative applications are available to use these devices to understand patient ailments and monitor them better. The prediction is made with the help of the AT-DLM technique to yield valuable results and proper analytics of the datasets under consideration. The dressing of the data is done at the initial level to ensure that the data is free from redundancy, re-occurrences, miscellaneous information, missing information, excellent values, appropriate inputs, and any interference. A more precise method for data dressing is done to ensure that there is no record in the dataset which make it redundant or duplicated. The availability of data in a clear and precise format is helpful for the proper execution of the algorithm for disease prediction. There are several layers in which the system proposed during this study works. The first layer contains data dressing. At the next level the data is analyzed. In the next stage the data analyzed ensures that the results are clear and complete. The next layer deals with the update of the data to a secure blockchain based component. The final layer ensures the privacy and final commit of the dataset in the blockchain. It comprises several hidden layers along with enhanced and modified mechanisms for optimization of the monitoring of the healthcare system. The collection of the data, along with the preparation as well as assessment, is done in three different phases. The sub-layers for prediction and performance analysis are done at another stage after the collection and dressing of the data is completed. Yet another layer is responsible for handling the data provided in a raw state. The processing layer is

accountable for cleaning and preparing the data to improve the consistency between the data and remove any inconsistencies. From the above deliberation, it can be concluded that there exists a need for:

- Implementation of AT-DLM algorithm for identification of information monitoring at the client side.
- Integration of an algorithm to capture the information at the patient level and identify the severity of the disease.
- Integration of an intelligent system for managing and detecting intrusion to ensure data flow from the healthcare unit to the centralized blockchain network.
- Implement an enhanced and more authentic IoMT system to ensure sustainability between higher medical standards.
- Use of a federated learning environment, which will ensure the safety and secrecy of the information from an individual perspective.
- Ensure the testing of the model, which is trained with the help of a real-time dataset collected by a small healthcare unit at the local level. The output of the model is evaluated at the final global model level.
- To ensure that humanity is getting proper benefits related to diagnostics of information and prediction of diseases.

The model defined in this study is executed in the following sequence:

1. Initially, the hospitals identify all the components and the devices that are connected to the monitoring of the healthcare parameters for a patient. The information collected by the IoMT devices and systems are uploaded to a centralized local server at the healthcare facility.
2. The data is collected and appropriately gathered for further analysis at the local level. The preprocessing of the data takes place at the initial layer in the local storage.
3. The data collected from the devices may contain errors or noise that need to be addressed and cleaned before any specific processing takes place to generate the local model with the help of machine learning algorithms.
4. The preprocessing of the data leads to the normalization of the information and the removal of any noise that can arrive in the data set collected by the IoMT system.
5. The application layer sends the data from the initial unit to the final processing layer. At this layer, the prediction of the information, along with the diagnostics, takes place.

The model proposed in this study collects data from the IoMT devices implanted in the hospitals. Patient monitoring is done using these devices and data is collected by these small sensor based IoMT devices. Before submission to other layer, the data is further processed and finally sent to the next level. This layer is responsible for the cleaning of data and making it suitable for complex prediction and diagnosis. Once the diagnosis is completed along with the data collected from the IoMT devices, information is further submitted to the blockchain for further commit. This data is completely accepted by the blockchain after complete verification of the data sender authenticity. If the data is transmitted by the legitimate user, then the data is further processed in the blockchain. The final commit is completed after the verification of user and exchange of the smart contract between the

parties involved. Thus the complete model proposed will produce value added data in a secure manner for global usage.

## SIMULATION RESULTS

In this study, we used the Parkinson's disease dataset for testing the model. The model used the FL approach coupled with the dataset for intrusion detection. The data collected from the IoMT unit was processed correctly and accumulated at the local storage level. A random dataset classification was done with 75% of the data for training purposes and 25% for testing purposes. In this approach we analyzed the data on the client and the server side. The model was designed to collect the data from the client side. The collected data at the client location was transformed into a common format and supplied to the server side for validation. Once the data was collected, the complete information should be submitted to the validation side. The server side validated the data to ensure that the information retrieved from the client was integral and authentic. The blockchain secure digital exchange made sure that the client was a valid agency supplying the information. Once the information was collected, the validation took place. The hash functions for the blockchain ensured that the data was collected without any loss and that intrusion had not taken place in the transfer procedure.

Table 2 presents a comparison of our proposed model with previous models based on various parameters such as the purpose of the model (prediction, diagnosis), and technologies embedded in the model (such as fused, blockchain, IDS, FL). Previous models shown in Table 2 above use a unified database system in which the information for data privacy or information leakage is widespread. FL provided a well-known solution to this problem in which the information collected from confidential sources was not revealed to the network or transmitted across the nodes. The decentralized architecture made it possible to hold sensitive data using a private framework for the healthcare units.

The efficiency of the proposed framework was measured using the true positive rate(TPR), true negative rate (TNR), positive prediction value (PPV), and negative prediction value (NPV). These were calculated as per the equations suggested by *Yamada et al. (2019)*.

$$TPR = \frac{TP}{(TP + FN)} \tag{1}$$

$$TNR = \frac{TN}{(TN + FP)} \tag{2}$$

$$PPV = \frac{TP}{(TP + FP)} \tag{3}$$

$$NPV = \frac{TN}{(TN + FN)}. \tag{4}$$

The dataset was split into approximately 400 records on the client side during the training. These records were divided into attacker records as well as regular records. The forecasting system received great accuracy for the intrusion at the client side, as per

**Table 2 Comparison of our proposed model with other state-of-the-art models.**

| Study | Data | Prediction | Diagnosis | Fused | Healthcare | Blockchain | IDS | FL |
|---|---|---|---|---|---|---|---|---|
| *Ihnaini et al. (2021)* AI | Medical | Y | Y | N | N | Y | N | N |
| *Dang et al. (2022)* IoT Fusion | Medical | Y | Y | Y | N | N | N | N |
| *Gadekallu et al. (2020)* Blockchain | Medical | Y | Y | N | N | N | N | N |
| *Xu et al. (2021)* Sensors | Medical | N | N | N | N | Y | N | N |
| *Li et al. (2021)* Edge/Fog | Medical | Y | Y | Y | N | N | N | N |
| *Adeniyi, Ogundokun & Awotunde (2021)* IoT IDS | Medical | Y | Y | Y | N | N | N | N |
| *Sedik et al. (2020)* FL Model | Medical | Y | Y | N | N | Y | N | Y |
| Our Study | Medical | Y | Y | Y | Y | Y | Y | Y |

**Table 3 Performance evaluation during training for the proposed architecture—client side.**

| Location | Sensitivity | Accuracy | Specificity | FN | FD | FP | NPV |
|---|---|---|---|---|---|---|---|
| C1 | 0.9812 | 0.9325 | 0.8315 | 0.0177 | 0.0700 | 0.1755 | 0.9766 |
| C2 | 0.9719 | 0.9522 | 0.8517 | 0.0106 | 0.0612 | 0.1579 | 0.9614 |
| C3 | 0.9789 | 0.9517 | 0.9222 | 0.0109 | 0.0213 | 0.1946 | 0.9312 |
| C4 | 0.9899 | 0.9796 | 0.8763 | 0.0173 | 0.0357 | 0.1325 | 0.9433 |

**Table 4 Performance evaluation during validation for the proposed architecture—client side.**

| Location | Sensitivity | Accuracy | Specificity | FN | FD | FP | NPV |
|---|---|---|---|---|---|---|---|
| C1 | 0.9889 | 0.9664 | 0.8778 | 0.0317 | 0.0319 | 0.1097 | 0.9359 |
| C2 | 0.9387 | 0.9431 | 0.8698 | 0.0228 | 0.0337 | 0.1179 | 0.8957 |
| C3 | 0.9645 | 0.9579 | 0.9165 | 0.0158 | 0.0473 | 0.0913 | 0.9746 |
| C4 | 0.9421 | 0.9325 | 0.8532 | 0.0536 | 0.0391 | 0.1346 | 0.9533 |

the observations in Tables 3, 4 and 5. The system was adequate for accurately identifying intrusion at the client level, along with statistical information calculated during the training. The division of the records on the client side took place at four different levels. Several parameters, such as the miss rate, accuracy, specificity, and sensitivity of the data, were calculated with the help of the following equations, as suggested by *Al-Qarafi et al. (2022)*.

$$\text{Missrate} = \frac{\sum_{b=0}^{2}\left(Q_b/S_{z \neq b}\right)}{\sum_{b=0}^{2}(T_b)}, \quad \text{Where } z = 0, 1 \tag{5}$$

**Table 5  Performance evaluation with FL during server-side validation.**

| Location | Sensitivity | Accuracy | Specificity | FN | FD | FP | NPV |
|---|---|---|---|---|---|---|---|
| C1 | 0.9613 | 0.9324 | 0.7364 | 0.0346 | 0.0654 | 0.2721 | 0.8456 |
| C2 | 0.9712 | 0.9233 | 0.7755 | 0.0263 | 0.0256 | 0.2432 | 0.8389 |
| C3 | 0.9525 | 0.9696 | 0.8657 | 0.0269 | 0.0324 | 0.1787 | 0.8732 |
| C4 | 0.9736 | 0.9517 | 0.8159 | 0.0487 | 0.0863 | 0.1946 | 0.8431 |
| FL Server | 0.9879 | 0.9715 | 0.9046 | 0.0122 | 0.0198 | 0.0955 | 0.9054 |

$$\text{Accuracy} = \frac{\sum_{b=0}^{2}(Q_b/S_b)}{\sum_{b=0}^{2}(Q_b)} \tag{6}$$

$$\text{Specificity} = \frac{Q_0/S_0}{(Q_0/S_0 + Q_0/S_1)} \tag{7}$$

$$\text{Sensitivity} = \frac{Q_1/V_1}{(Q_1/S_0 + Q_{1/S_1})}. \tag{8}$$

$Q$ is the predictive output, $S$ is the actual output, $Q_0$ and $S_0$ represent that there is no disease detected, $Q_1$ and $S_1$ represent that intrusion was caught in the observation, $Q_b$ and $S_b$ represent the predictive and the actual values for the results from the system.

Table 3 presents information collected from four client locations: C1, C2, C3, and C4. These small healthcare units represent the source of data collected from smart devices and the IoMT system sensors. The records were taken from the Parkinson's disease dataset for evaluation. The forecasting system achieved a successful prediction rate per the observations. The statistical measurements were satisfactory for the method proposed in this study.

It is evident that client location C1 provided a sensitivity of 98.14%, specificity of 83.15%, accuracy of 93.25%, false-negative values of 1.77%, false positive value of 17.55%, false discovery rate of 7% and negative prediction value for 97%. For the client C2 location, the observation reveals a sensitivity of 97%, accuracy of 95.22%, specificity of 85.17%, false-negative value of 1.06%, false discovery for 6.12%, false positive values of 15.79% and finally negative predictive values for 96.14% is achieved. Various client location C3 yields the productive values for a sensitivity of 97.89%, accuracy of 95.17%, specificity of 92.22%, false-negative values of 1.09%, false discovery of 2.13%, false positive value for 19.46% and finally the negative prediction value for 93.12%. In contrast, for the client location C4, the sensitivity is 98.99%, accuracy is 97.96%, specificity is 87.63%, false-negative values for 1.73%, false discovery for 3.57%, false positive for 13.25% and negative prediction value for 94.33% is received. Figure 5 illustrates the results of observations that were taken to evaluate the performance during the training of the proposed architecture at the client site.

The intrusion detection and its validation are done on the client side for the proposed architecture in this study. The parameters for the performance evaluation are calculated with Eqs. (1)–(8). The dataset comprises 400 records, split to measure the training and the validation. All the statistical measurements prove that the model is eligible for acceptance in the fifth generation of healthcare systems.

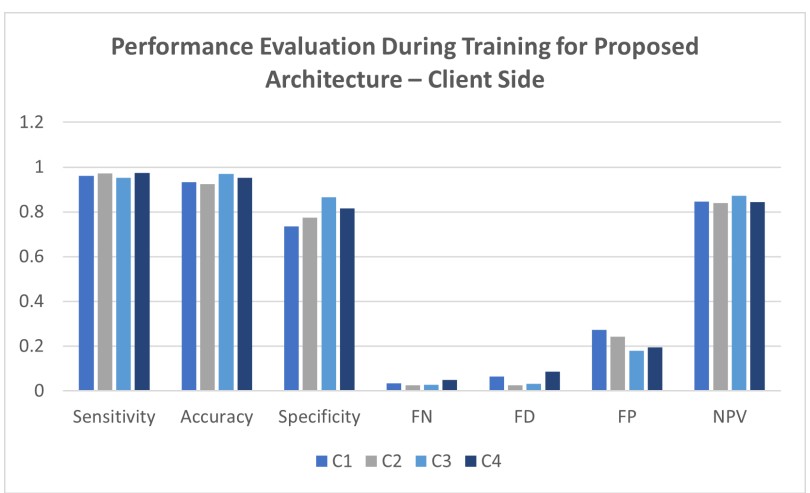

**Figure 5** **Performance evaluation during training for the proposed architecture—client side.**

Table 4 represents the performance evaluation for the proposed architecture to predict the intrusion during the validation of the system. The validation process was done for a total of around 250 records. These records were validated at variable client sites. The attack records were successfully separated from the normal records to validate the model's accuracy during the validation process. Several statistical methods were taken into consideration for validating every client site individually. Several client locations give excellent results during the validation process with the attacking and standard records. Client C1 was able to achieve a sensitivity of 98.89%, accuracy of 96.64%, specificity of 87.78%, false-negative values of 3.17%, false discovery rate of 3.19%, false positive rate of 10.97% and negative predictive value for 93.59%. In comparison, client C2 achieved a sensitivity of 93.87%, accuracy of 94.31%, specificity of 86.98%, false-negative value of 2.28%, false discovery rate of 3.37%, false positives for 11.79% and negative prediction value for 89.57%. The third client location for C3 produced a sensitivity of 96.45%, an accuracy of 95.79%, specificity of 91.65%, false-negative values of 1.58%, false discovery for 4.73%, false positive for 9.13% and negative prediction value for 97.46%. At the same time, the final point C4 site gave a sensitivity of 94.21%, accuracy of 93.25%, specificity of 85.32%, false-negative value of 5.36%, false discovery of 3.91%, false positive of 13.46% and negative prediction value of 95.33% was observed. A comparative graph for the observations is placed in Fig. 6, reflecting the same results.

The performance evaluation during the validation process gave fantastic results per the expectation for the proposed architecture in this study. It was indeed required that the assessment of the model during the training and validation process be cross-verified at the server-side validation.

Table 5 presents values for evaluating the performance at the server side of the system where FL was applied. The training rules from various client site locations were collected and sent to the server side. The dataset was evaluated on the server side, comprising all the training rules. These rules were submitted after the training and validation of the

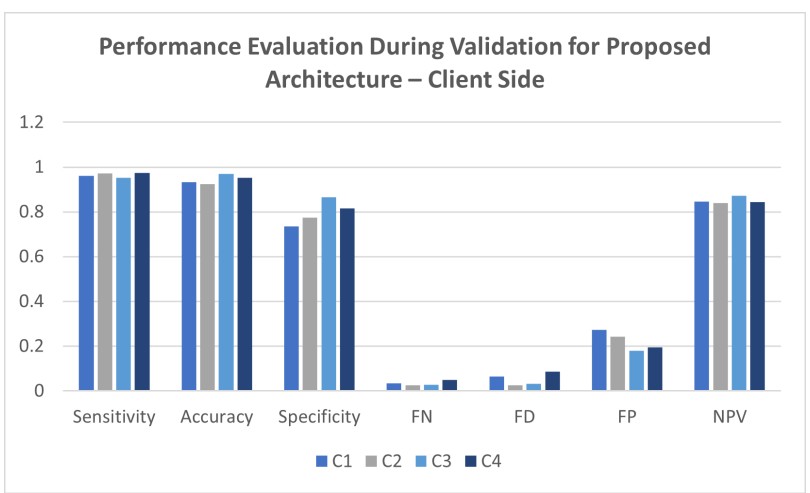

**Figure 6** Performance evaluation during validation for proposed architecture—client side.

individual models were done at client locations. The results are mentioned in the table
for various client locations and the FL-oriented server site. Similar rules were applied for
the machine learning-based methodology at the client locations and the server side. The
C1 client achieved a sensitivity of 96.13%, an accuracy of 93.24%, a specificity of 73.64%,
a false-negative rate of 3.46%, a false discovery of 6.54%, a false positive rate of 27.21%,
and negative predictive value of 84.56%. In contrast, client C2 was able to achieve 97.12%
sensitivity, accuracy of 92.33%, specificity of 77.55%, false-negative value of 2.63%, a
false discovery rate of 2.56%, false positive rate of 24.32%, and negative predictive value
of 83.89%. The client C3 site location achieved 95.25% sensitivity, accuracy of 96.96%,
specificity of 86.57%, false-negative rate of 2.69%, false discovery of 3.24%, false positive
rate of 17.87% and negative prediction value of 87.32%. Client C4 achieved sensitivity
of 97.36%, accuracy of 95.17%, specificity of 81.59%, false-negative value of 4.87%, false
discovery of 8.63%, false positive rate of 19.46%, and negative predictive value of 84.31%.
The observations for the client side were under the validation and training evaluation
of the proposed model. However, the FL server site yielded 98.79% sensitivity, 97.15%
accuracy, 90.46% specificity, 1.22% false-negative values, 1.98% fake discovery, 9.5% false
positive value and negative prediction value of 90.54%. The overall performance of the
FL integration at the server side was satisfactory and better than the previously available
models, as shown in Fig. 7.

From the experimental results, it was observed that various models have been proposed
to achieve a higher rate of prediction of Parkinson's disease. These machine learning-based
models imply different techniques with variable outputs and results. *Qi et al. (2020)*
presented a convolutional neural network (CNN) based model to be used for prediction
that achieved an accuracy rate of 84.5%. The tenfold cross-validation proposed by *Adeniyi,
Ogundokun & Awotunde (2021)* achieved 90.6% accuracy. The results of our proposed
model is better when compared with the use of CNN. An artificial neural network along
with lean year and RBF SVM was applied by *Nakamoto (2008)*, which resulted in 89.3%

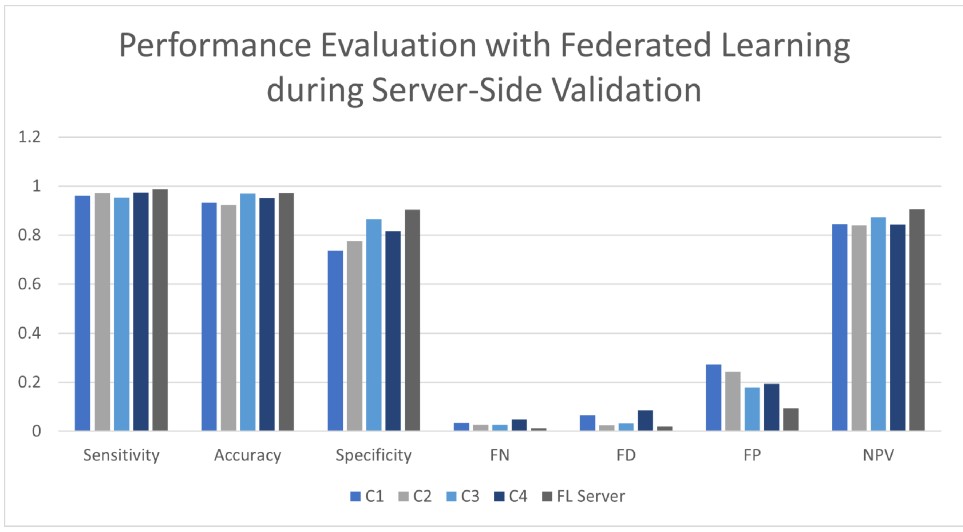

**Figure 7  Performance evaluation with FL during server-side validation.**

prediction accuracy. The use of logistic regressions and the random forest algorithm was done by *Medjahed et al. (2011)* to provide an accuracy of 90.1%. The integration of SVM along with the tenfold cross-validation was done by *Yu et al. (2015)* to give an accuracy of 91.25%. Our proposed proposed in this study uses federated learning, which is integrated with the intrusion detection model for the fifth generation of healthcare, and yielded an accuracy level of 97.13%. This level of accuracy is better than previous published work and would be beneficial for achieving a secure modern healthcare system.

## CONCLUSION

The adoption of intelligent healthcare facilities based on the use of IoMT devices is gaining popularity. The clinical data collected for disease prediction is of immense importance for medical diagnostics and treatment. Since the amount of raw data is increasing rapidly, proper information must be extracted to provide accurate diagnosis and medical advice. Patient care, along with the privacy of the data, are two critical factors that should be taken into consideration during the design of any model used for disease prediction related to the healthcare units. AI and machine learning algorithms empower the system to provide fruitful results from existing datasets. Integrating machine learning rules is one of the best practices AI can provide to healthcare units. However, the collection of bulk data creates risks related to cyber-attacks, malicious activities, and potential hacks. It is a necessary to detect intrusions in such circumstances to ensure provenance of data. The model proposed in this study includes an intrusion detection component. Blockchain is being widely used where data is critical and privacy of information is a significant concern. Adopting Federated learning and the blockchain enhances the predictive power of the IoMT system proposed in this study. Various frameworks have been analyzed and released recently to identify the use of clinical data for disease prediction. The accuracy achieved

in this study is better than previously published models and reasonably acceptable. The primary identification of the information and the result from our study is up to the mark. The proposed framework in this research is helpful for the fifth generation of healthcare; however, with the growth of hidden layers, it is expected that greater precision should be applied to reduce the processing complexity in the system. The system's performance with various configurations on the client and the server side will be a future area of development.

### Funding

This work was supported by the Deanship for Research Innovation, Ministry of Education in Saudi Arabia through the project number IFP22UQU4400260DSR231. The funders had no role in study design, data collection and analysis, decision to publish, or preparation of the manuscript.

### Grant Disclosures

The following grant information was disclosed by the authors:
Deanship for Research Innovation, Ministry of Education in Saudi Arabia through the project number: IFP22UQU4400260DSR231.

### Competing Interests

The authors declare there are no competing interests.

### Author Contributions

- Jameel Almalki conceived and designed the experiments, analyzed the data, authored or reviewed drafts of the article, and approved the final draft.
- Saeed M. Alshahrani performed the experiments, authored or reviewed drafts of the article, and approved the final draft.
- Nayyar Ahmed Khan conceived and designed the experiments, performed the computation work, prepared figures and/or tables, and approved the final draft.

### Data Availability

The Parkinson's Disease Data Set is available in the Supplemental File and at Kaggle: https://www.kaggle.com/datasets/vikasukani/parkinsons-disease-data-set.

### Supplemental Information

Supplemental information for this article can be found online at http://dx.doi.org/10.7717/peerj-cs.1778#supplemental-information.

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
