# Peer review of "A comprehensive secure system enabling healthcare 5.0 using federated learning, intrusion detection and blockchain"

_PeerJ Computer Science, doi:10.7717/peerj-cs.1778_

## Round 0.1 · original submission · Major Revisions

Reviewers find merit in this paper and recommend it for revision. Authors are required to thoroughly read the reviewers' comments/suggestions, and revise the manuscript accordingly. The revised manuscript will be subjected to re-review by the reviewers.

**Language Note:** The review process has identified that the English language must be improved. PeerJ can provide language editing services - please contact us at copyediting@peerj.com for pricing (be sure to provide your manuscript number and title). Alternatively, you should make your own arrangements to improve the language quality and provide details in your response letter. – PeerJ Staff

Reviewer 1 ·

Basic reporting

a. It is highly recommended that the English of the paper should be properly checked and updated.
b. It is also recommended to complete the Proof Reading for the paper by some proofreading(editing) agency / fluent English speaker to make sure that the English used is in proper format (especially the grammar and vocabulary).
c. The abstract of the paper should be condensed strictly accordingly to the topic of the paper.

Experimental design

a. Include citation to latest published literature.

Validity of the findings

a. The comparison should be in association with the context. Try to modify the comparisons and present them as needed.

·

Basic reporting

The article represented by the authors is of potential importance in relevance with the topic presented. Slight modifications are expected as below:
Language check for the paper is expected and should be given special concern.
The conclusion of the paper must be explained more.
The paper should include more modern literature / citations as per the topic relevance.

Experimental design

The results must be explained somewhat in accordance with the paper.
The methodology must explain blockchain in a little detail as per the use in the paper / model suggested.

Validity of the findings

Authors are advised to check the quality of the figures as per the journal guidelines.

Additional comments

Updates relative to the above suggestions will fulfill the requirements for the paper to be published.

Reviewer 3 ·

Basic reporting

no comment

Experimental design

no comment

Validity of the findings

no comment

Additional comments

The topic addressed in the manuscript is potentially interesting and the manuscript contains some ‎practical meanings, however, there are some issues, which should be addressed by the authors to ‎enhance further the article:‎
‎- In the first place, I would encourage the authors to complete the Introduction Section more with the ‎key contributions. The Introduction section can be made much more impressive by highlighting the ‎main contributions. The contribution of the study should be explained simply and clearly. In addition, ‎the motivation should be stated more clearly.‎
‎- The abstract is too long. The authors should summarize the results and contents of the abstract of the ‎manuscript. Some content you can move them to the introduction section.‎
‎- The readability and presentation of the study should be further improved. The paper suffers from ‎language problems.‎
‎- The “Related Work” section needs a major revision in terms of providing a more accurate and ‎informative literature review and the pros and cons of the available approaches and how the proposed ‎method is different comparatively. Here are some papers: “Performance Modeling and Analysis of IoT-‎enabled Healthcare Monitoring Systems”, “A secure healthcare 5.0 system based on blockchain ‎technology entangled with federated learning technique”, “User privacy prevention model using ‎supervised federated learning‐based block chain approach for internet of Medical Things”, ‎‎“Performance Evaluation of IoT-Fog-Cloud Deployment for Healthcare Services”, etc. ‎
‎- The importance of the architecture carried out in this manuscript can be explained better, especially ‎the role of the Blockchain in the proposed architecture.‎
‎- Simulation and Results Section should be edited in a more highlighting, argumentative way. The ‎author should analyze the reason why the tested results are achieved on the Client side and validation ‎side.‎
‎- How to set the parameters of the proposed architecture for better performance?‎
‎- It will be helpful to the readers if some discussions about the insight of the main results are added as ‎recommendations or use cases.

---

## Round 0.2 · Minor Revisions

Dear Authors,
One of the reviewers has suggested minor revisions. You are required to consider all the comments and suggestions of the reviewer and revise your manuscript accordingly. Highlight all the changes and upload a response to the reviewer's comment. Looking forward to receiving your revised manuscript. Good luck.

Reviewer 1 ·

Basic reporting

The required corrections have been made well.

Experimental design

The required corrections have been made well.

Validity of the findings

The required corrections have been made well.

Additional comments

The required corrections have been made well.

Reviewer 3 ·

Basic reporting

.

Experimental design

.

Validity of the findings

.

Additional comments

The authors have meticulously revised their manuscript. All my reviews have been answered. The overall quality of the article and its contributions have improved significantly. It is recommended that the article be accepted for publication.

Reviewer 4 ·

Basic reporting

- This manuscript is easy to follow and self-explanatory in the present form.

- All comments are well-addressed by the authors.

- This manuscript can be accept in the present form.

Experimental design

Experiments evaluation results are significantly better.

Validity of the findings

Acceptable

Additional comments

Accept

·

Basic reporting

The introductory section provides a general overview of the Internet of Medical Things (IoMT), its growing popularity, and challenges related to security and privacy​​. However, it might benefit from a more detailed historical context and specific examples illustrating the evolution and current state of IoMT. This would enhance the readers' understanding of the topic and its relevance.

Experimental design

The research question appears to be well-defined and relevant, focusing on addressing security issues in smart healthcare through Blockchain and federated learning​​. The methods described, including the use of the Parkinson’s dataset and the NSL-KDD dataset, seem appropriate for the research objectives​​. However, there is room for more explicit detailing of the procedures and protocols followed in the study to enable replication and assessment by other researchers.

Validity of the findings

The findings, including the proposed model’s high accuracy in disease analysis and intrusion detection, are promising​​. However, the article might benefit from a more rigorous statistical analysis of the results. Additionally, comparisons with existing models or systems in similar domains could provide a clearer understanding of the impact and novelty of the study​​. The findings, including the proposed model’s high accuracy in disease analysis and intrusion detection, are promising​​. However, the article might benefit from a more rigorous statistical analysis of the results. Additionally, comparisons with existing models or systems in similar domains could provide a clearer understanding of the impact and novelty of the study​​.

Additional comments

Various components like the blockchain, Actual Time – Deep Learning Model (AT-DLM), and federated learning are discussed with technical details​​. However, a more comprehensive explanation of how these components interact and complement each other in the proposed system would be beneficial. Also, elaborating on the specific innovations or improvements each component brings to the IoMT field would add value. The article covers data amalgamation and intrusion detection as critical aspects of the system​​. Further clarification on how these processes are specifically tailored or optimized for the healthcare context could enhance the article's relevance and applicability.

---

## Round 0.3 · accepted · Accept

The authors have addressed the minor comments raised by one of the reviewers. The manuscript may be accepted in its current form.